# Design of a Charge Pump Circuit and System with Input Impedance Modulation for a Flexible-Type Thermoelectric Generator with High-Output Impedance

**Kazuma Koketsu and Toru Tanzawa \***

Graduate School of Integrated Science and Technology, Shizuoka University, Hamamatsu 432-8561, Japan;
kohketsu.kazuma.15@shizuoka.ac.jp
\* Correspondence: toru.tanzawa@shizuoka.ac.jp

**Abstract:** This paper describes a charge pump system for a flexible thermoelectric generator (TEG). Even though the TEG has high-output impedance, the system controls the input voltage to keep it higher than the minimum operating voltage by modulating the input impedance of the charge pump using two-phase operation with low- and high-input impedance modes. The average input impedance can be matched with the output impedance of the TEG. How the system can be designed is also described in detail. A design demonstration was performed for the TEG with 400 Ω. The fabricated system was also measured with a flexible-type TEG based on carbon nanotubes. Even with an output impedance of 1.4 kΩ, the system converted thermal energy into electric power of 30 μW at 2.5 V to the following sensor ICs.

**Keywords:** charge pump; energy harvesting; thermoelectric; IoT

## 1. Introduction

The Internet of Things (IoT) currently is attracting researchers' attention, which is a system for the interaction of information from things such as sensing edge devices to the cloud and servers via the Internet and vice versa [1]. The maintenance costs to replace batteries can be a large portion of the costs of edge devices. Therefore, it is expected that sensing devices should be battery free based on the energy transducer generating electric power from environmental energy such as sunlight and vibration kinetic energy. A thermoelectric generator (TEG) extracts power from a temperature gradient. The open-circuit voltage $V_{OC}$ of the TEG increases in proportion to the temperature difference between hot and cold heat sources [2]. Bulk-type TEGs [3] have a low output impedance ($R_{TEG}$) of the order of Ω and are in production together with boost converters. Flexible-type thin film TEGs [4] are expected to have various applications because they can be placed on curved surfaces. A drawback of the flexible-type TEG is the high-output impedance of the order of 10–100 Ω, especially in the case of a small form-factor. Even worse, a low-cost small form-factor TEG generates $V_{OC}$ as low as a few hundred mV. To operate sensor ICs, boost converters are required [5–7]. In this research, the design of boost charge pump circuits (CPs) is proposed for a flexible-type TEG with high-output impedance, as illustrated in Figure 1. Such a system is used for heat pipes [8] and wrist watches [9].

To design systems with TEGs and integrated CPs, the circuit area and power conversion efficiency (PCE) are key figures of merit. Table 1 summarizes the key features of existing designs and this work. In [10], the design of low-voltage CPs was developed to strike a balance between the circuit area and power efficiency under the conditions of a given output voltage and current. In this design, CPs are driven by voltage sources with zero impedance, while TEGs have a finite output impedance. In [11], both TEGs and CPs were optimally designed to minimize their areas when CPs were driven by TEGs. However, design constraints such as temperature differences and the number of TEG units connected

in parallel and in series were not taken into consideration. A design methodology was proposed when $V_{OC}$ and $R_{TEG}$ were given in [12,13]. In [12], an optimum design was provided to determine the dimensions of switching devices and the clock frequency to maximize the output power of the CP when the number of stages $N$ and stage capacitors $C$ of the CP and the $V_{OC}$ and $R_{TEG}$ of TEG were given. However, the output voltage of the CP was not given, whereas the input voltage of the load circuit must be controlled with a specific voltage. In [13], how the input voltage of the CP or the output voltage of the TEG is determined theoretically was discussed when the circuit area of the CP was minimized or, in other words, when the output power of the CP was maximized with a given CP circuit area to generate a target output current at a specific output voltage, as shown in Figure 2a, which is the same target design of this work. However, the minimum operation voltage of circuits was not considered in [13], but it was assumed that the input voltage of the CP can be set at any voltage. Furthermore, no control circuit was disclosed to control the input voltage of the CP in [13]. In this work, the minimum operating voltage of the circuits was taken into consideration in the design, as shown in Figure 2b. This can be a key design point especially for TEGs with a high-output impedance, which have a potentially large IR drop at $V_{DD}$.

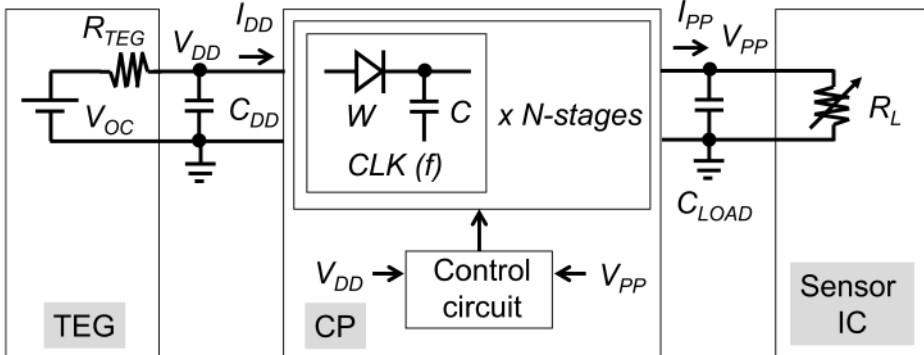

**Figure 1.** Block diagram of the energy harvesting system based on the TEG and CP.

**Table 1.** Comparison of the key features of this work with existing designs.

| | Optimum Design Target | Given Design Parameters | Parameter to Be Optimized | Parameters to Be Determined |
|---|---|---|---|---|
| Tokuda [10] | CP | $V_{PP}, I_{PP}, V_{DD}, f$ | Area of the CP to be minimized and the PCE to be maximized | $N, C$ |
| Koketsu [11] | TEG + CP | $V_{PP}, I_{PP}, f$ | Area of the TEG and CP to be minimized | $V_{OC}, R_{TEG}, N, C$ |
| Lu [12] | | $V_{OC}, R_{TEG}, N, C$ | $P_{PP}$ to be maximized | $W, f$ |
| Tanzawa [13] | | $V_{OC}, R_{TEG}, V_{PP}, f$ | $I_{PP}$ @ $V_{PP}$ to be maximized | $N, C$ |
| This work | | $V_{OC}, R_{TEG}, V_{PP}, f, V_{DD}{}^{MIN}$ | | |

　　　　This paper is an extended version of a conference paper [14] to describe its details. A control circuit to operate the CP was proposed to meet the demand that the output current be generated as high as the target current at a specific voltage while the input voltage of the CP is controlled at a voltage higher than the minimum operating voltage. The designs of the CP system and building blocks are presented in Sections 2.1 and 2.2, respectively, to discuss how the circuits can be optimally designed. The entire system was fabricated in 65 nm CMOS. Experimental results are shown in Section 2.3, and Section 3 gives a summary of this work.

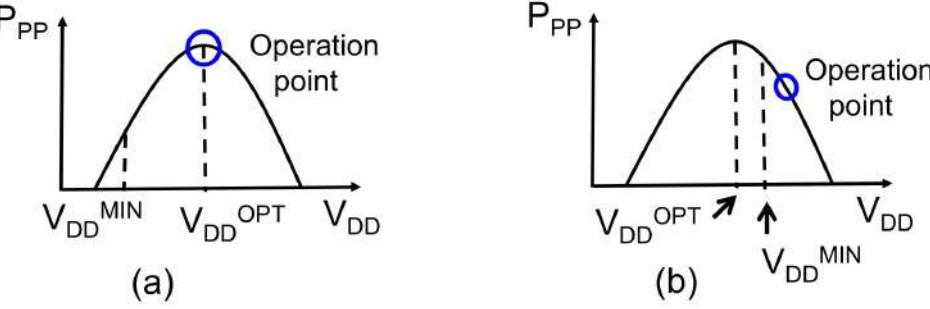

**Figure 2.** Operating points of [8] (**a**) and of this work (**b**).

## 2. Circuit Design

### 2.1. System Design

Figure 3 illustrates the proposed CP system to extract power from the TEG with high-output impedance and to drive the following sensor ICs. Table 2 shows the condition to resume or suspend CP operation. A detector DETi monitors $V_{DD}$ and outputs ENi. A detector DETo monitors $V_{PP}$ and outputs ENo. Only when both signals become high, an oscillator OSC outputs a clock to drive the CP. Otherwise, the OSC stops working to not drive the CP. The third detector DETpp generates a signal VPP_OK to let the sensor ICs know the supply voltage is sufficiently high to work.

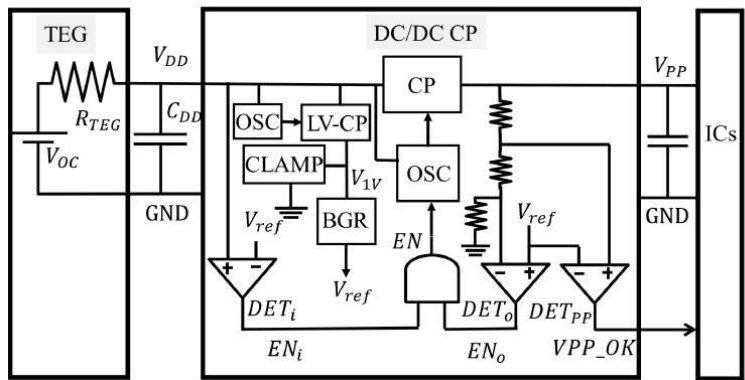

**Figure 3.** Building blocks of the proposed CP circuit system.

**Table 2.** Operation condition of the main CP against $V_{DD}$ and $V_{PP}$.

|  | $V_{PP} < V_{PPT}$ | $V_{PP} > V_{PPT}$ |
|---|---|---|
| $V_{DD} > V_{DDT}$ | Resume | Suspend |
| $V_{DD} < V_{DDT}$ | Suspend | Suspend |

Figure 4 shows two operation phases in steady state. In Phase (a), the CP inputs the current mainly from $C_{DD}$. Even though $R_{TEG}$ is much larger than the input impedance of CP, $V_{DD}$ can be controlled to be higher than $V_{DD\_MIN}$. Phase (a) starts with EN high when $V_{PP}$ hits $V_{PPM} = V_{PPT}$, where $V_{PPM}$ and $V_{PPT}$ are the minimum voltage of $V_{PP}$ and the target voltage of $V_{PP}$, respectively. $V_{PP}$ increases while $V_{DD}$ decreases due to CP operation. EN goes low when (1) ENo goes low or (2) ENi goes low. In the case of (1), the ripple $\triangle V_{PP}$ is determined by the loop response from the output node of the CP to EN. $V_{DDM}$ must be higher than $V_{DDT}$. In the case of (2), $V_{DDM}$ is equal to $V_{DDT}$. In Phase (b), $V_{DD}$ increases with the charging current from the TEG, while $V_{PP}$ decreases with the discharging load current. The input impedance of the CP becomes very large because the main charge pump CP is suspended with EN low, even though a small amount of current flows into small building blocks such as LV-CP. Thus, even though the TEG has high-output impedance, the system controls the input voltage to keep it higher than the minimum operating voltage by

modulating the input impedance of the charge pump using two-phase operation with low- and high-input impedance modes. The average input impedance can be matched with the output impedance of the TEG. On the other hand, such an operation is not required when the output impedance of TEGs such as the bulk-type is much lower than the input impedance of the CP in operation. The operating point approaches $V_{OC}$, but the system can work as long as $V_{OC}$ is higher than the minimum operating voltage.

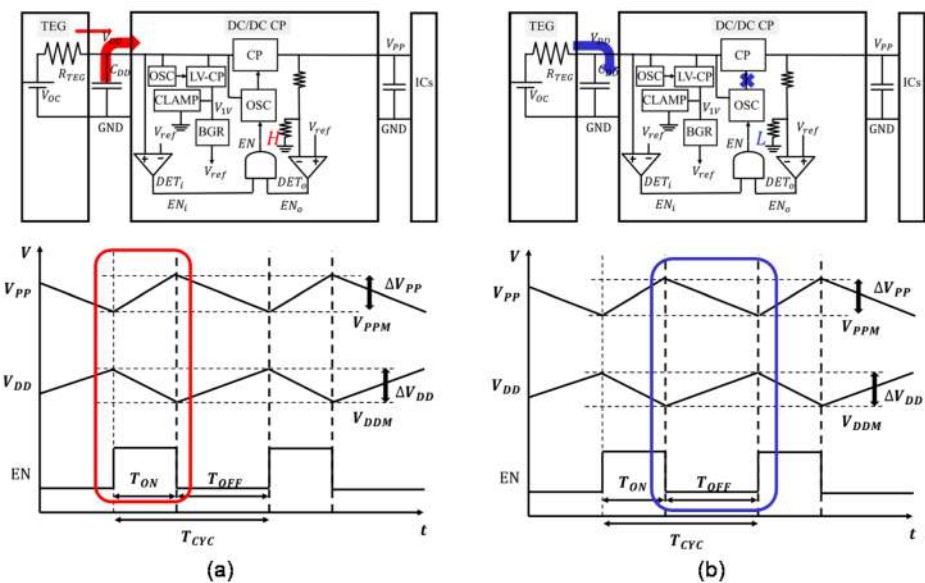

**Figure 4.** Two phases of the circuit operation. (**a**) Low and (**b**) high input impedance modes

The following equations hold among $T_{ON}$, $T_{OFF}$, $\Delta V_{PP}$ and $\Delta V_{DD}$, where it is assumed that $I_{PP}$ and $I_{DD}$ are the steady-state currents and can be treated as constant when $\Delta V_{PP} << V_{PP}$, $\Delta V_{DD} << V_{DD}$, and $I_{TEG} << I_{DD}$.

$$T_{ON} = \frac{C_{DD}\,\Delta V_{DD}}{I_{DD}} = \frac{C_{PP}\,\Delta V_{PP}}{I_{PP} - I_{LOAD}} \tag{1}$$

$$T_{OFF} = R_{TEG}C_{DD}\ln\frac{V_{OC} - V_{DDM}}{V_{OC} - V_{DDM} - \Delta V_{DD}} = \frac{C_{PP}\,\Delta V_{PP}}{I_{LOAD}} \tag{2}$$

$I_{PP}$ and $I_{LOAD}$ are related as Equation (3).

$$I_{LOAD} = \frac{T_{ON}}{T}I_{PP} \tag{3}$$

When one can regard $I_{TEG}$ as constant in the case of $\Delta V_{DD} << V_{DD}$, $I_{DD}$ and $I_{TEG}$ are related as Equation (4).

$$I_{TEG} = \frac{T_{ON}}{T}I_{DD} \tag{4}$$

### 2.2. Building Blocks' Design

B1: Main charge pump

The given design parameters are the minimum open-circuit voltage of the TEG ($V_{OCMIN}$), $R_{TEG}$, $V_{PPT}$. The number of stages $N$ was designed to maximize $I_{PP}$ at $V_{PPT}$ when the circuit area is given. Based on [15], $N$ is given by Equation (5).

$$N = [1.7 \times N_{MIN}] = \left\lceil 1.7 \times \frac{V_{PP} - V_{DD} + V_{TH}^{EFF}}{V_{DD}/(1 + \alpha_T) - V_{TH}^{EFF}} \right\rceil, \tag{5}$$

where $[x]$ indicates the floor function of $x$, $N_{MIN}$ is the minimum number of stages to barely generate $V_{PP}$, and $V_{TH}^{EFF}$ is an effective threshold voltage of switching transistors, which were called ultra-low power diodes in [16]. The capacitance of each stage capacitor $C$ is related with $I_{PP}$ and $I_{DD}$ as Equations (6) and (7), where the clock frequency $f$ is determined to maximize $I_{PP}$.

$$I_{PP} = \frac{fC(1+\alpha_T)}{N}\left[\left(\frac{N}{1+\alpha_T}+1\right)V_{DD} - (N+1)V_{TH}^{EFF} - V_{PP}\right], \tag{6}$$

$$I_{DD} = \left(\frac{N}{1+\alpha_T}+1\right)I_{PP} + \left(\frac{\alpha_T}{1+\alpha_T}+\alpha_B\right)fNCV_{DD} + I_{CTRL}, \tag{7}$$

where $\alpha_T$ and $\alpha_B$ are the ratios of the top ($C_{TOP}$) and bottom plate parasitic capacitance ($C_{BTM}$) to $C$, $C_{TOP}/C$ and $C_{BTM}/C$, respectively. Note that $C_{BTM}$ includes the parasitic capacitance of an oscillator to drive the main CP. $I_{CTRL}$ is the input current for the control circuits, which was assumed to be $\beta I_{DD}$ using the design parameter $\beta$ (<1) in this paper because the auxiliary circuits assumed in this paper as shown later steadily ran regardless of $T_{ON}$. $I_{DD}$ is also given by Equation (8) at the extreme case of $T_{ON} = T$ and $T_{OFF} = 0$.

$$I_{DD} = \frac{V_{OC} - V_{DD}}{R_{TEG}} \tag{8}$$

From Equations (6)–(8), the minimum C needs to meet Equation (9).

$$C = \frac{(1-\beta)N(V_{OC} - V_{DD})}{fR_{TEG}\left[(N+1+\alpha_T)\left\{\left(\frac{N}{1+\alpha_T}+1\right)V_{DD} - (N+1)V_{TH}^{EFF} - V_{PP}\right\} + \left(\frac{\alpha_T}{1+\alpha_T}+\alpha_B\right)N^2V_{DD}\right]} \tag{9}$$

To have a duty ratio of $T_{ON}/T$ smaller than a factor of $\gamma$, the C to be designed must be increased by a factor of $1/\gamma$.

The parameters shown in Table 3 were used for design demonstration. $V_{DDMIN}$ was mainly determined by the technology used to design, e.g., the availability of low-Vt CMOS and circuits used in the system. As will be shown later, it was limited by an oscillator to generate a clock with 10 MHz. Such a moderate frequency was required to have a sufficiently small circuit system built in the same sensor ICs. From Equations (5) and (9), N and C were calculated to be 19 and 4.8 pF at $V_{DDT} = 0.5$ V, respectively. Figure 5 shows $P_{PP}$ and CP area $NC/\gamma$ as a function of $V_{DD}$.

**Table 3.** Design parameters used in this work.

| $V_{DDMIN}$ | $V_{DDT}$ | $V_{TH}^{EFF}$ | $V_{PPT}$ | $R_{TEG}$ | $V_{OCMIN}$ | $f$ | $\alpha_T$ | $\alpha_B$ | $\beta$ | $\gamma$ |
|---|---|---|---|---|---|---|---|---|---|---|
| 0.45 V | 0.5 V | 0.25 V | 2.5 V | 400 Ω | 0.6 V | 10 MHz | 0.1 | 0.2 | 0.2 | 0.33 |

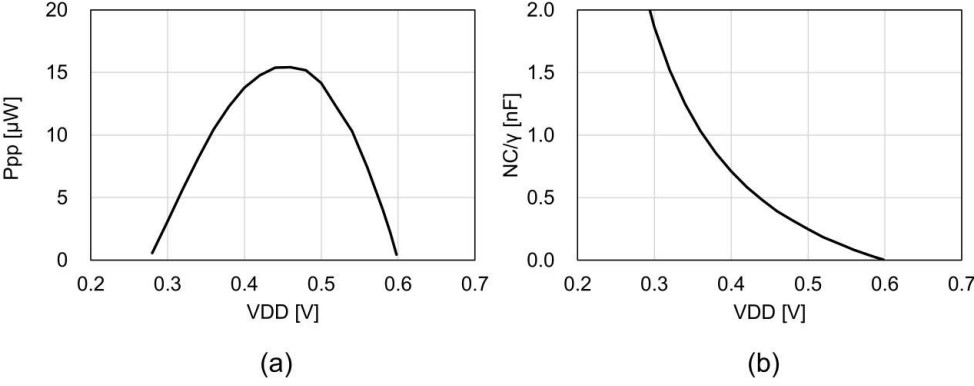

**Figure 5.** (a) $P_{PP}$ vs. $V_{DD}$; (b) CP area $NC/\gamma$ vs. $V_{DD}$.

B2: Auxiliary circuits

As illustrated in Figure 3, the detectors compare $V_{DD}$ and $V_{PP}$ with a reference voltage $V_{REF}$ generated by bandgap reference BGR [17]. To provide a supply voltage $V_{1V} \sim 1$ V to the BGR, another small CP (LV-CP) was implemented. The LV-CP is operated in open loop not to affect the $V_{DDMIN}$ of the system. A dedicated oscillator starts running without any input signal other than $V_{DD}$. When LV-CP converts power to the output terminal and $V_{1V}$ reaches about 1 V, a clamping circuit CLAMP with NMOSFETs connected in series with the output terminal clamps the output voltage. $V_{1V}$ is also used as the supply voltage of all the logic gates and the detectors. Figure 6 shows a simulated result of the BGR. $V_{REF}$ is saturated when $V_{1V} > 0.8$ V.

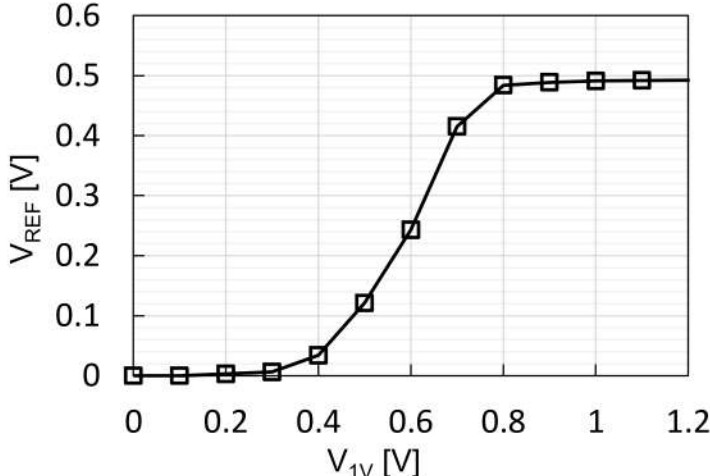

**Figure 6.** $V_{REF}$ vs. $V_{DD}$.

### 2.3. Experimental Results

The system was designed in 65 nm low-Vt CMOS technology, as shown in Figure 7. The entire area was 0.28 μm². The CPs had an *N* of 20 and a *C* of 15 pF. The LV-CP had an *N* of six and a *C* of 3 pF to generate the supply current of 10 μA at 1 V, which was sufficiently high for the following circuits while keeping $\gamma < 0.2$.

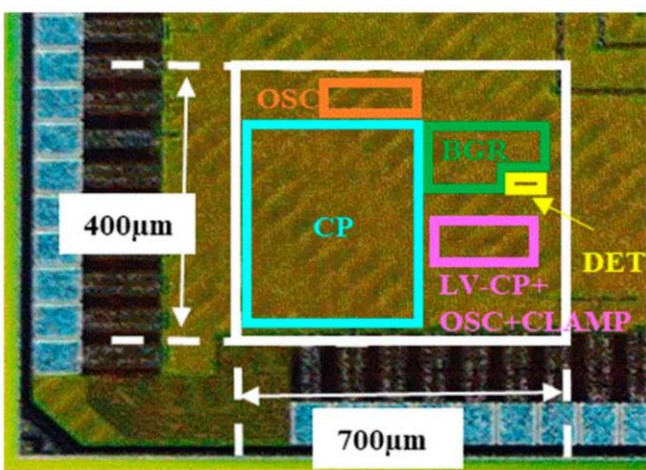

**Figure 7.** Die photo.

The input terminal was connected to an equivalent circuit of the TEG with $V_{OC}$ and $R_{TEG}$. A $C_{DD}$ of 300 nF and a $C_{PP}$ of 1 nF were connected to the input and output terminals of the CP system, respectively. Since the system did not work at a $V_{OC}$ of 0.6 V probably because the $V_{TH}$ of MOSFETs was close to the slow corner while the simulation was performed at the typical corner, the experiments were performed at $V_{OC}$ of 0.8 V. Figure 8

shows $I_{PP}$, $I_{DD}$, $V_{DD}$, and $P_{PP}$ as a function of $V_{PP}$ where $V_{PP}$ was varied by varying the load resistance. All the simulations were performed with the slow-corner model. The measured results were matched with the simulated ones with an error of about 10%. $V_{PP}$ was regulated at 2.5 V when $I_{PP}$ was 25 μA or lower. The average $V_{DD}$ was 0.6 V or higher when $V_{PP}$ was regulated.

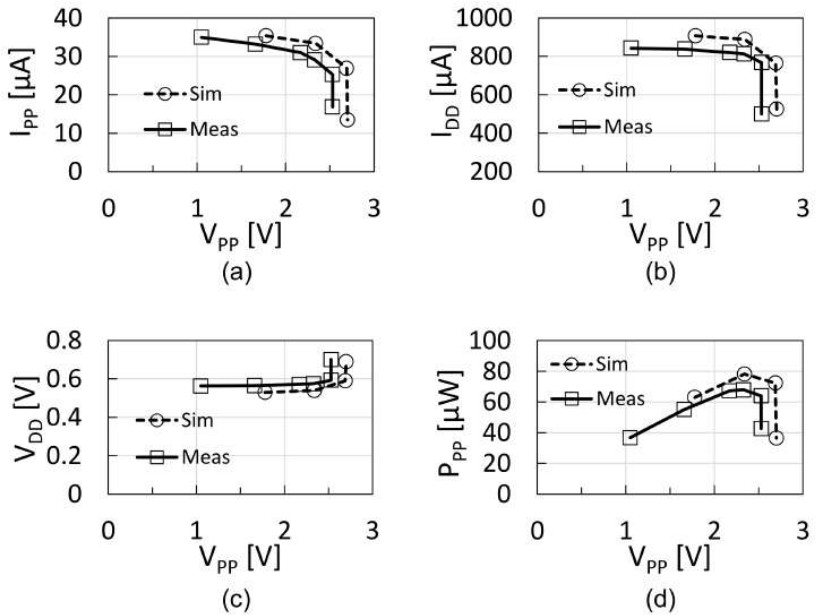

**Figure 8.** (**a**) $I_{PP}$, (**b**) $I_{DD}$, (**c**) $V_{DD}$, and (**d**) $P_{PP}$ as a function of $V_{PP}$.

To see the dynamic response of $V_{PP}$ and $V_{DD}$ against $V_{OC}$, $V_{OC}$ was made to go up and down between 0.5 V and 1 V in 200 μs, as shown in Figure 9. A signal EN was also monitored using a buffer whose supply voltage was $V_{1V}$. In the period $T_1$, because $V_{DD}$ was lower than $V_{DDT}$, EN stayed low. In the period $T_2$, because $V_{DD}$ was higher than $V_{DDT}$, but $V_{PP}$ was lower than $V_{PPT}$, EN stayed high. Once $V_{PP}$ reached $V_{PPT}$, in the period $T_3$, the system stayed in the steady state where the $T_{ON}/T_{OFF}$ operation was repeated to keep $V_{PP}$ and $V_{DD}$ at $V_{PPT}$ and $V_{DDT}$, respectively.

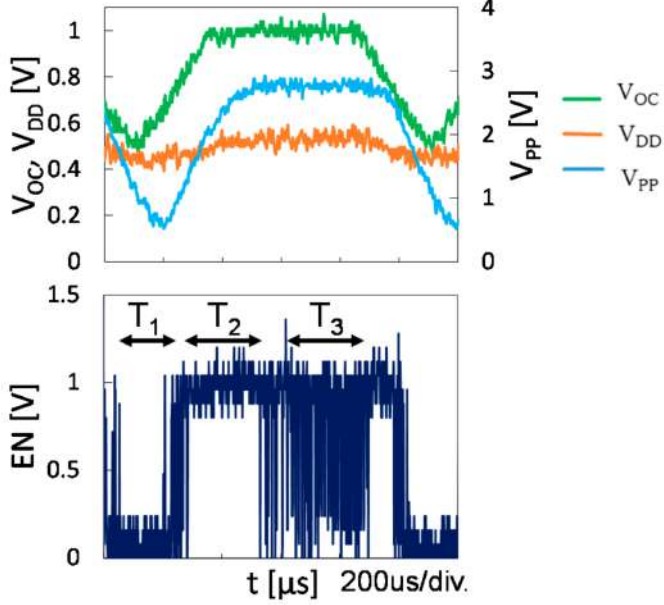

**Figure 9.** Dynamic behavior of $V_{PP}$ and $V_{DD}$ against $V_{OC}$.

The system was also tested with the TEG using a thermal source, as shown in Figure 10. The TEG was based on carbon nanotubes [18]. The TEG module was built to fit with a pipe, which flowed hot liquid or gas. Because the TEG module had an $R_{TEG}$ of 1.4 kΩ, $V_{OC}$ needed to be set at a higher voltage of 1.1 V with a temperature difference of 66 K to enable the fabricated converter system to be functional, as shown in Figure 11.

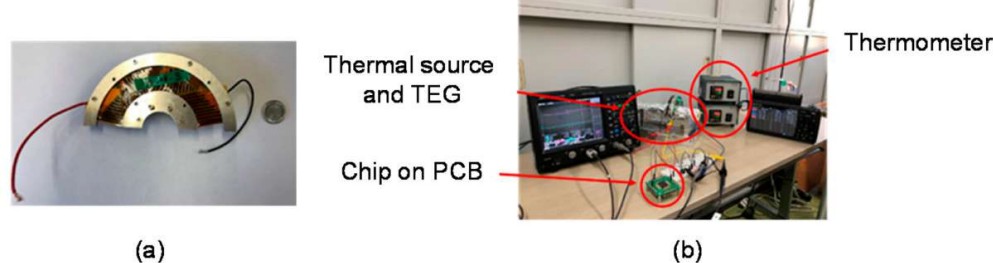

**Figure 10. The** TEG module (**a**) and experimental setup with the TEG (**b**).

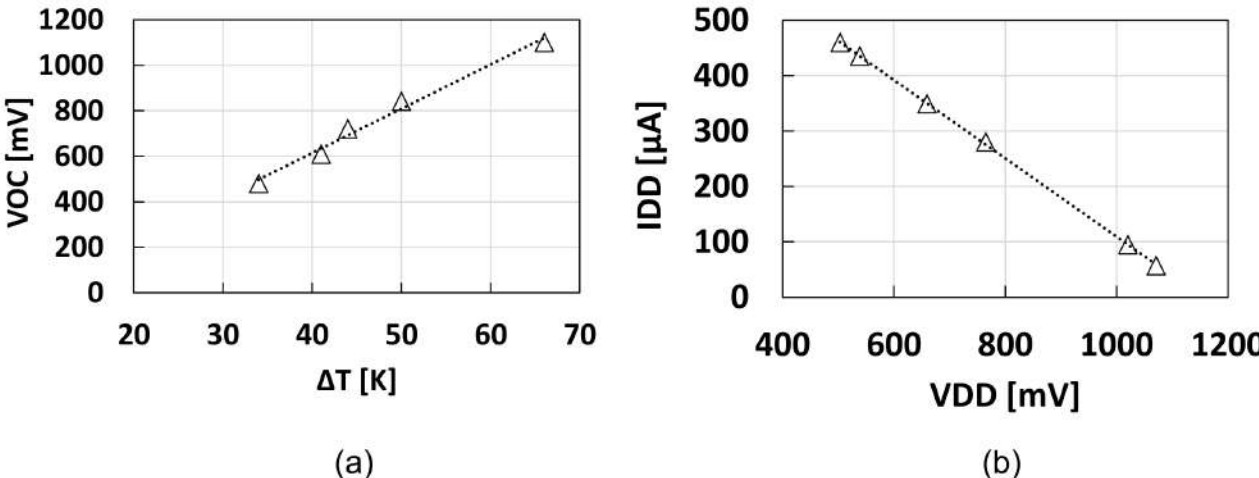

**Figure 11.** $V_{OC}$ vs. $\Delta T$ (**a**) and $I_{DD}$ vs. $V_{DD}$ at $\Delta T$ of 66 K (**b**).

Figure 12 shows $I_{PP}$, $I_{DD}$, $V_{DD}$, $P_{PP}$, $\eta_{SYS}$, and $\eta_{CP}$ as a function of $V_{PP}$. $\eta_{SYS}$ and $\eta_{CP}$ are defined by $(V_{PP} \times I_{PP})/(V_{OC} \times I_{DD})$ and $(V_{PP} \times I_{PP})/(V_{DD} \times I_{DD})$, respectively. The $V_{PP}$ regulation point was different by 0.3 V between measured and simulated, but the electric values except for it were in good agreement. It was confirmed that the converter system with the TEG module under the experimental condition could supply power of 30 µW at 2.5 V to the following sensor ICs. The overall power conversion efficiency $\eta_{SYS}$ was hit at about 7% against a theoretical limit with no loss of 50%. The power conversion efficiency of the converter system $\eta_{CP}$ was 15% when $V_{DD}$ was 0.55 V at $V_{PP}$ of 2.5 V, i.e., a voltage ratio ($V_{PP}/V_{DD}$) of 4.5. For comparison, $\eta_{CP}$ of 20%, 32%, and 45% was realized with a $V_{DD}$ of 0.1 V, 0.2 V, and 0.3 V at a $V_{PP}$ of 0.5 V, respectively, in [10]. Thus, the $\eta_{CP}$ of the proposed converter system was a little lower than that of [10] at the voltage ratio of 4.5. The design optimization may need to be improved to increase power conversion efficiency by including the TEG electrical parameters in the design parameters.

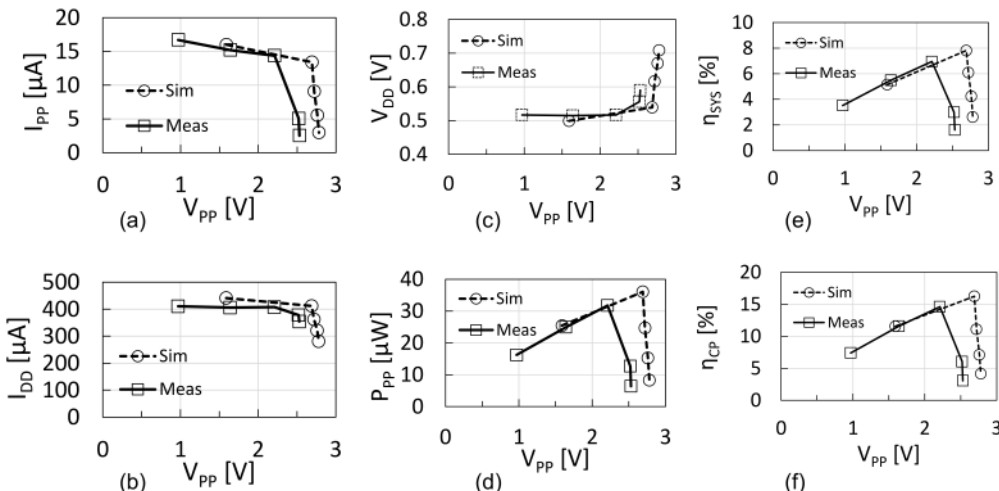

**Figure 12.** (**a**) $I_{PP}$, (**b**) $I_{DD}$, (**c**) $V_{DD}$, (**d**) $P_{PP}$, (**e**) $\eta_{SYS}$, and (**f**) $\eta_{CP}$ as a function of $V_{PP}$.

### 3. Conclusions

A charge pump circuit system was presented for energy harvesting based on a flexible-type thermoelectric generator with high-output impedance. Even though the charge pump was operated with a highly resistive TEG, the input voltage could be controlled at a voltage higher than $V_{DDMIN}$ by modulating the input impedance of the CP using two-phase operation with low- and high-input impedance modes. The average input impedance could be matched with the output impedance of TEG. The design methodology was proposed to determine the $N$ and $C$ of the main charge pump when $V_{OC}$, $R_{TEG}$, $V_{PP}$, $f$, and $V_{DD}^{MIN}$ were given. The system was fabricated in 65 nm CMOS to demonstrate the functionality of the system with the TEG. Using an equivalent circuit for the TEG, the system was validated with a $V_{OC}$ of 0.8 V and an $R_{TEG}$ of 400 Ω. $V_{PP}$ regulation was successfully observed. The circuit system was also measured with a flexible-type TEG and a thermal source. The system converted thermal energy into power to 30 µW at 2.5 V. By adding a full-bridge rectifier between the energy transducer and the proposed converter, the control circuit would be able to work even with other energy transducers such as piezoelectric or electrostatic vibration energy transducers with an AC equivalent voltage source and high-output impedance.

**Author Contributions:** Conceptualization, T.T.; methodology, K.K. and T.T.; software, K.K.; validation, K.K. and T.T.; formal analysis, K.K. and T.T.; investigation, K.K. and T.T.; writing—original draft preparation, K.K.; writing—review and editing, T.T.; funding acquisition, T.T. All authors have read and agreed to the published version of the manuscript.

**Funding:** This research was partially funded by Zeon Corp. and Micron Foundation.

**Acknowledgments:** This work was supported by Zeon Corp., VDEC, Synopsys, Inc., Cadence Design Systems, Inc. Rohm Corp., and Micron Foundation. The authors wish to thank M. Futagawa and H. Hirano and S. Ota for technical discussions.

**Conflicts of Interest:** The authors declare no conflict of interest.

### Nomenclature

| | |
|---|---|
| C | Capacitance per stage |
| $C_{DD}$ | Capacitor connected to $V_{DD}$ |
| $C_{PP}$ | Capacitor connected to $V_{PP}$ |
| f | Clock frequency |
| $I_{CP}$ | Input current of CP |
| $I_{CTRL}$ | Input current of control circuits |

| | |
|---|---|
| $I_{DD}$ | Operating current of the TEG and CP |
| $I_{LOAD}$ | Load current of the CP |
| $I_{TEG}$ | Output current of the TEG |
| N | Stage number of the CP |
| $P_{DD}$ | Input power of the CP |
| $P_{PP}$ | Output power of the CP |
| $P_{TEG}$ | Generated power of the TEG |
| $\alpha_T$ | Ratio of top plate capacitance to $C$ |
| $\alpha_B$ | Ratio of bottom plate capacitance to $C$ |
| $R_{TEG}$ | Output impedance of the TEG |
| T | Operation period, $T_{ON} + T_{OFF}$ |
| $T_{OFF}$ | Suspended period |
| $T_{ON}$ | Resumed period |
| $V_{DD}$ | Input voltage of the CP |
| $V_{DDT}$ | Target input voltage of the CP to be controlled |
| $V_{DDM}$ | Minimum $V_{DD}$ in operation |
| $\Delta V_{DD}$ | Ripple in $V_{DD}$ |
| $V_{PP}$ | Output voltage of the CP |
| $V_{PPT}$ | Target output voltage of the CP to be controlled |
| $V_{PPM}$ | Minimum $V_{PP}$ in operation |
| $\Delta V_{PP}$ | Ripple in $V_{PP}$ |
| $\beta$ | Ratio of $I_{CTRL}$ to $I_{DD}$ |
| $\gamma$ | Operation duty; $T_{ON}/T$ |

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
