# Peer review of "Design of a Charge Pump Circuit and System with Input Impedance Modulation for a Flexible-Type Thermoelectric Generator with High-Output Impedance"

_electronics, doi:10.3390/electronics10101212_

Round 1

Reviewer 1 Report

This manuscript presents optimization of charge pump circuit design for thermoelectric generators. Unlikely a previous conference paper, the manuscript mainly targets maximizing Ipp at Vppt point. This work technically sounds. 

However, the reviewer raises major concern that the optimized results such as the number of stages and capacitance are not used to design the system. Rather than using the parameters from optimization, they are the same as the parameters of the conference paper. Eventually, "Circuit design" chapter and "Experiments" chapter are not integrated well. The reviewer gently recommends the authors integrate both sections well.

Minor comment: Fig.8 (d) y-axis has the wrong unit. The author should depict power using uW, not uA.

Author Response

Thanks for your feedback. Please see the attachment.

Reviewer 2 Report

I think this is a high-quality manuscript. One question, is there any way to further improve the power conversion efficiency?

Author Response

(The authors gave the same response as above.)

Reviewer 3 Report

The manuscript describes the design, implementation and experimental characterization of a charge pump system for energy extraction from a flexible thermoelectric generator. The flexible TEG based on carbon nanotube is not described. The manuscript is well organized and includes a detailed introduction describing the state-of-the-art and motivation of the work. The section on circuit design is well elaborated. The system concept and building blocks are sufficiently supported by theoretical considerations. 

The following is a list of comments for further improvement of the paper:

The charge pump was designed in particular for a flexible TEG with high output impedance. The reader might be interested in what the advantage of this particular TEG is and how the system compares to other systems with conventional, not flexible TEG.

Line 177: the overall power conversion efficiency is denoted with about 7%. Does it include the efficiency of the TEG as well? What is the efficiency of the circuit itself? How does the proposed charge pump compare to other charge pumps in terms of efficiency?

Can the proposed concept of the charge pump circuit also be used for other energy converters with high output impedance such as piezoelectric or electrostatic transducers?

Author Response

(The authors gave the same response as above.)
